# HIRA stabilizes skeletal muscle lineage identity

Joana Esteves de Lima [1], Reem Bou Akar[1,4], Léo Machado[1,4], Yuefeng Li[2,3,4], Bernadette Drayton-Libotte[1],
F. Jeffrey Dilworth [2,3] & Frédéric Relaix [1✉]

The epigenetic mechanisms coordinating the maintenance of adult cellular lineages and the inhibition of alternative cell fates remain poorly understood. Here we show that targeted ablation of the histone chaperone HIRA in myogenic cells leads to extensive transcriptional modifications, consistent with a role in maintaining skeletal muscle cellular identity. We demonstrate that conditional ablation of HIRA in muscle stem cells of adult mice compromises their capacity to regenerate and self-renew, leading to tissue repair failure. Chromatin analysis of *Hira*-deficient cells show a significant reduction of histone variant H3.3 deposition and H3K27ac modification at regulatory regions of muscle genes. Additionally, we find that genes from alternative lineages are ectopically expressed in *Hira*-mutant cells via MLL1/MLL2-mediated increase of H3K4me3 mark at silent promoter regions. Therefore, we conclude that HIRA sustains the chromatin landscape governing muscle cell lineage identity via incorporation of H3.3 at muscle gene regulatory regions, while preventing the expression of alternative lineage genes.

[1] Univ Paris Est Creteil, INSERM, EnvA, EFS, AP-HP, IMRB, F-94010 Creteil, France. [2] Department of Cellular and Molecular Medicine, University of Ottawa, Ottawa, ON, Canada. [3] Sprott Centre for Stem Cell Research, Ottawa Hospital Research Institute, Ottawa, ON, Canada. [4] These authors contributed equally: Reem Bou Akar, Léo Machado, Yuefeng Li. ✉email: frederic.relaix@inserm.fr

Skeletal muscle has a high regeneration potential that relies on tissue-resident muscle stem cells, also known as satellite cells. These adult stem cells were identified nearly 60 years ago as quiescent cells lying on the surface of the myofiber underneath the basal lamina[1]. Muscle stem cells express the paired-box transcription factor PAX7 which is required for their maintenance and myogenic lineage progression[2]. In resting muscles, quiescent satellite cells become activated upon injury to repair damaged tissue, while a fraction self-renews to regain quiescence. Activated satellite cells express MYF5 and MYOD and re-enter the cell cycle, proliferating as myoblasts that will either fuse to generate newly formed fibers or return to quiescence to restore the muscle stem cell pool[3]. Proper regulation of quiescence, activation and differentiation of muscle stem cells ensures the normal homeostasis of the tissue as well as its repair in the case of disease or injury-related damage. Although the transcriptional regulation of these processes is widely studied, the molecular mechanisms regulating PAX7 expression and maintaining muscle stem cell identity have not been identified.

Epigenetic regulation of gene expression is crucial for cell fate commitment and maintenance. The epigenetic memory of a specified cell defines the state of gene expression that is transmitted upon division in order to sustain cell lineage identity[4]. This mechanism of inheritance of a gene expression pattern relies on the epigenetic signature associated with DNA methylation, chromatin architecture, histone variants deposition, and histone modifications[4,5]. Nucleosome incorporation of the non-canonical histone H3 variant H3.3 has been associated with epigenetic memory and is required to maintain the transcriptionally active pattern of a tissue[6]. The H3.3 histone is specifically incorporated into the nucleosomes by two histone chaperones, HIRA and DAXX, depending on the genomic loci[7,8]. In mouse embryonic stem cells (ESCs), HIRA regulates H3.3 deposition at transcription start sites (TSS), promoters, and gene bodies while DAXX deposits H3.3 at the telomeres and constitutive heterochromatin[7,8]. H3.3 deposition is not only required for pluripotency maintenance but also for somatic cell gene expression regulation, including the neurogenic and myogenic lineages, playing a central role in cellular differentiation[9–12]. While it has been previously shown that HIRA and H3.3 are regulating myogenic cell differentiation[9–11] their role in regulating PAX7 and sustaining lineage identity has not been addressed.

Here, we show that HIRA is a major upstream regulator of muscle stem cells and myogenic lineage identity. Conditional knockout of HIRA in muscle stem cells leads to defective skeletal muscle regeneration in vivo. We further demonstrate that HIRA-deficient cells lose PAX7 and muscle-specific gene expression, while inducing tissue-specific genes from other lineages. We show that HIRA drives myogenic identity by specifically incorporating H3.3 at regulatory regions of muscle genes. By contrast, induction of non-myogenic genes in HIRA-deficient muscle cells requires MLL1/MLL2-mediated deposition of H3K4me3 mark at silent or bivalent loci. Altogether, our data demonstrate that HIRA is required to maintain muscle stem cell lineage chromatin landscape in order to promote the expression of myogenic-specific genes and prevent the expression of alternative lineages.

## Results

**HIRA sustains myogenic cell identity**. In order to identify upstream epigenetic regulators of the muscle stem cell-specific transcription factor PAX7, we performed targeted deletion of candidate genes in the myoblast C2C12 cell line. Among these, we addressed the role of HIRA as a potential upstream regulator of PAX7 by mutating Exon 10 of *Hira* using Crispr/Cas9

(Supplementary Fig. 1a–c), to generate a truncated protein lacking the putative H3.3 interaction domain[13]. The deletion of *Hira* did not affect the expression of *Daxx*/DAXX (Supplementary Fig. 1b, c). Upon *Hira* ablation, expression of PAX7 was drastically reduced and MYOD protein levels decreased (Fig. 1a–c). In addition, *Hira* mutant myogenic cells displayed impaired differentiation in low-serum-containing medium as seen with the quantification of Myosin-positive cells (Fig. 1d, e), consistent with previous observations[11]. *Hira*-deficient cells did not show changes in proliferation or cell death (Supplementary Fig. 1d–g). We analyzed two other independent *Hira*-KO clones for muscle phenotype by RT-qPCR and the loss of PAX7 and myogenic gene expression was consistent between them (Supplementary Fig. 1h). To characterize the global impact of the loss of *Hira* we performed RNA-sequencing (RNA-seq) in proliferating *Hira*-KO and control C2C12 cells cultured in high serum-containing medium and at low density (Fig. 1f, g). Strikingly, we observed major changes in the transcriptome of *Hira* KO compared to control cells with 30% of expressed genes being dysregulated (10297 out of 33686 detected genes), suggesting a global impact on gene expression in the absence of HIRA (Fig. 1h). This was unexpected given that *Hira* deletion in mouse ESCs does not result in major transcriptomic changes[8,14]. Gene ontology and analysis of downregulated genes, included genes linked with muscle stem cell lineage, including muscle stem cell markers, muscle lineage progression, and differentiation (Fig. 1i, k, l). Surprisingly, upregulated transcripts were related to non-myogenic lineages, such as vascular or neuronal lineages (Fig. 1j). Moreover, we found various endothelial, mesenchymal, neuronal, adipogenic, and osteogenic lineage-specific genes to be ectopically upregulated in myogenic cells when *Hira* was deleted (Fig. 1k, l). In addition, Hoxb cluster genes, previously identified as repressed in C2C12 cells[15], showed a strong upregulation in the *Hira*-KO cell line (Fig. 1k, l). Taken together, these results show that HIRA is required to sustain muscle stem cells and myogenic gene expression and to inhibit the expression of genes from alternative lineages in C2C12 cells, making it an essential epigenetic factor required for the maintenance of myogenic cell identity at a transcriptomic level.

**H3.3 deposition by HIRA maintains H3K27ac mark in myogenic gene loci**. We next evaluated whether the maintenance of skeletal muscle lineage identity is linked with the histone chaperone activity of HIRA and H3.3 incorporation. In order to determine if the deletion of *Hira* had an impact on the H3.3 incorporation in the genome, we performed H3.3 chromatin immunoprecipitation (ChIP) followed by sequencing (ChIP-seq) in proliferating *Hira*-deficient and control cells (cultured in high serum-containing medium and at low density). Genome-wide analysis showed a tenfold reduction in the H3.3 genomic deposition in the absence of *Hira* compared to control cells (38439 vs. 3744 detected peaks) (Fig. 2a). In addition, we observed that in myoblasts lacking *Hira* the number of H3.3 peaks is evenly decreased across genomic loci (Fig. 2b), suggesting a global pattern for H3.3 incorporation by HIRA in the genome of C2C12 cells, in contrast to findings in ESCs[8,14,16]. To investigate the H3.3 deposition pattern in muscle cells, we plotted the H3.3 average intensity signal along the promoter (±3 kb around the TSS), TSS and gene bodies in control and *Hira*-KO cells. We observed that H3.3 signal is diminished in *Hira*-KO cells when analyzing all protein-coding genes (Fig. 2c). To evaluate if decreased H3.3 deposition is linked with gene expression, we analyzed H3.3 average signal in upregulated or downregulated genes of the *Hira*-KO cell line compared with control cells. This analysis demonstrated that reduced H3.3 incorporation in

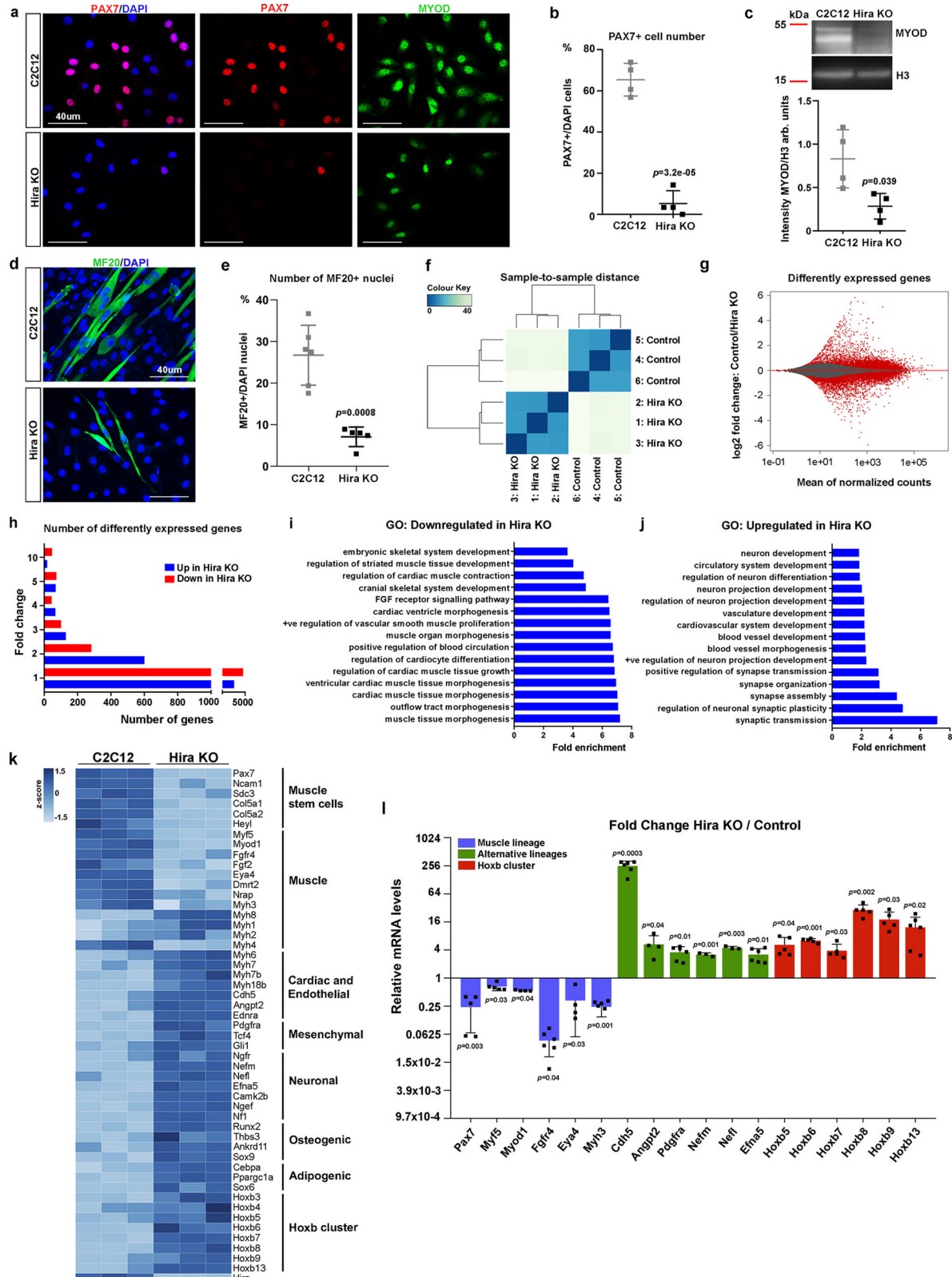

promoters and TSS is specifically associated with genes that are downregulated in *Hira* KO but not with genes that are upregulated, the latter displaying H3.3 levels similar to control (Fig. 2c). In addition, we confirmed that skeletal muscle genes (GO:0007519) in *Hira* KO follow the same pattern of H3.3 levels as the downregulated genes (Fig. 2c). Given that incorporation of H3.3 is required for acetylation and promoter-distal enhancer

(±3 kb around the TSS) activity in ESCs[14] we further analyzed the distribution of the acetylated histone H3 Lys27 (H3K27ac) that is associated with active enhancers and promoters[17], in *Hira*-deficient and control myoblasts using ChIP-seq. While H3K27ac modification was not altered in average when examined across all protein-coding genes, specific enrichment was observed in upregulated genes and conversely, downregulated genes showed

**Fig. 1 HIRA is required to maintain myogenic cell identity. a** Co-immunostaining with PAX7 (red) and MYOD (green) antibodies and the nuclear marker DAPI (blue) in C2C12 and *Hira*-KO cell lines. **b** Quantification of the number of PAX7-positive cells among DAPI-positive cells in **a** ($n = 4$ control and $n = 4$ cKO independent culture experiments). Error bars, mean ± SD, two-tailed unpaired *t*-test. **c** Western blot for MYOD and H3 (control) in C2C12 ($n = 4$ independent protein samples) and *Hira* KO ($n = 4$ independent protein samples). Uncropped blots in Source Data. Quantification of the signal intensity performed in four independent blots, comparing samples within each blot (bottom, arb. units: arbitrary units). Error bars, mean ± SD, two-tailed unpaired *t*-test. **d** Immunostaining with MF20 antibody (green) to visualize myosins and the nuclear marker DAPI (blue) in C2C12 and *Hira*-KO cell lines. **e** Quantification of the number of MF20-positive nuclei among DAPI-positive nuclei in **d** (C2C12 $n = 6$, *Hira* KO $n = 5$, independent differentiation assays). Error bars, mean ± SD, two-tailed unpaired *t*-test. **f** Differential expression analysis of the C2C12 ($n = 3$ independent RNA samples) and *Hira* KO ($n = 3$ independent RNA samples) RNA-seq samples. **g** MA-plot of C2C12 over *Hira*-KO RNA-seq data. Significantly dysregulated genes are highlighted in red (FDR < 0.05). **h** Number of upregulated (4920 genes, blue) and downregulated (5377 genes, red) genes in *Hira* KO compared to C2C12. **i, j** Gene ontology analysis for biological processes of the downregulated (**i**) and upregulated (**j**) genes. Selected enriched terms are presented according to the fold enrichment. **k** Heatmap with the normalized reads per gene of C2C12 and *Hira*-KO RNA-seq data in individual triplicates. **l** RT-qPCR analyses of the mRNA levels of selected genes from the C2C12 RNA-seq performed in control and *Hira*-KO cells ($n = 6$ for Fgfr4, Cdh5, Pdgfra, Efna5, Hoxb13; $n = 5$ Pax7, Myf5, Myh3, Hoxb5, Hoxb6, Hoxb7, Hoxb8, Hoxb9; $n = 4$ Myod1, Eya4, Angpt2; $n = 3$ Nefm, Nefl; independent RNA samples). For each gene, the mRNA levels of the control cells were normalized to 1. Muscle lineage (blue), alternative lineages (green), and Hoxb cluster (red). Error bars, mean ± SD, two-sided unpaired *t*-test. Scale bars, 40 μm.

decreased H3K27ac levels (Fig. 2d). This result suggests that loss of H3.3 incorporation in promoters and TSS of downregulated genes correlates with the loss of transcriptionally permissive H3K27ac mark (Fig. 2d). As actively expressed genes have been shown to reside within accessible chromatin, we performed an assay for transposase-accessible chromatin sequencing (ATAC-seq) to determine if the chromatin accessibility is modified in muscle genes when *Hira* is deleted in myogenic cells compared with controls. The chromatin accessibility was greatly decreased in *Hira* KO compared to control cells as shown by the reduced ATAC-seq peaks (Supplementary Fig. 2a, b). This decreased accessibility is likely due to the replacement of H3.3 by H3.1 upon loss of HIRA (Supplementary Fig. 2e, f). In addition, we observed that both H3.3 incorporation and H3K27ac histone modification were strongly reduced at sites where ATAC-seq peaks were lost in *Hira*-KO cells, demonstrating that H3.3 and H3K27ac modification are associated with open chromatin genomic regions (Fig. 2e, f). Since we detected a specific loss in the expression of myogenic-related genes in *Hira*-deficient cells, we further analyzed H3.3 enrichment at muscle-specific loci. The regulatory regions of *Pax7* have not been characterized, but putative enhancer regions have been identified by H3K27ac ChIP-seq in quiescent satellite cells[18]. We first scanned *Pax7* locus for H3.3 deposition and H3K27ac at opened chromatin sites in control cells. Strikingly, all detected regions presented a common pattern for H3.3, H3K27ac, and ATAC-seq peaks in myogenic cells and lost all of these marks in the absence of HIRA (Fig. 2g). We next analyzed the genomic regions of muscle stem cells and myogenic markers downregulated in *Hira*-deficient cells (Fig. 1k). These include the satellite cell markers *Pax7* (Fig. 2g), *Heyl* (Fig. 2h), *Col5a1* (Fig. 2i), *Sdc3* (Supplementary Fig. 2c), the previously identified regulatory regions of *Dmrt2* (−18 kb) (Fig. 2j), *Myod1* (−20 kb) (Fig. 2k), and *Fgfr4* (+19.2 kb) (Fig. 2l)[19–21], as well as *Eya4* (Fig. 2m) and *Fgf2* (Supplementary Fig. 2d). We observed that all these muscle stem cell and muscle-specific regulatory regions presented H3.3 incorporation associated with H3K27ac histone mark at open chromatin sites in C2C12 myogenic cells (Fig. 2g–m, Supplementary Fig. 2c, d). Consistently, H3.3 incorporation, H3K27ac modification, and ATAC-seq peaks are abrogated in *Hira*-deleted cells (Fig. 2g–m, Supplementary Fig. 2c, d). Combining the analysis of the RNA-seq, ChIP-seq, and ATAC-seq in control and *Hira*-KO cells we conclude that the incorporation of H3.3 by HIRA is required for chromatin accessibility and H3K27 histone acetylation at regulatory regions of PAX7 and myogenic-specific genes in order to maintain their expression in the muscle lineage.

***Hira*-deficient satellite cells lack regenerative capacity**. To demonstrate if HIRA is also required to preserve myogenic stem cell identity in vivo, we analyzed the consequences of ablating *Hira* in activated and proliferating satellite cells. To specifically delete *Hira* in adult muscle stem cells we inter-crossed *Pax7CreErt2* and *Hirafl/fl* mouse lines[22,23]. Skeletal muscle satellite cells from adult *Pax7CreErt2;Hirafl/fl* mice treated with tamoxifen diet for 2 weeks displayed a sharp decrease in *Hira* expression (80%) and HIRA protein was not detected compared with control *Pax7CreErt2;Hirafl/+* mice, without affecting DAXX (Supplementary Fig. 3a, b). Because muscle stem cells reside in a quiescence state in homeostasis, we performed injury experiments in the *Tibialis anterior* (TA) muscles using BaCl$_2$ injection (Fig. 3a) that efficiently triggers satellite cell activation and muscle regeneration[24]. We analyzed the impact on regeneration 7 days post injury (dpi), to allow amplification of the proliferating satellite cell pool after activation, and at 28 dpi when the regeneration is completed[3]. At 7 dpi the number of PAX7-positive cells was significantly reduced in the conditional knockout (cKO) mice when normalized to the fiber number (Fig. 3b, c). Since HIRA regulates PAX7 expression (Fig. 1k, l), we further used M-Cadherin as a satellite cell marker and confirmed that the number of M-Cadherin-associated nuclei is decreased in the absence of HIRA in satellite cells, 7 dpi (Fig. 3d, e). At 28 dpi, the number of PAX7-positive cells in the *Pax7CreErt2;Hirafl/fl* muscles was further reduced compared to 7 dpi (Fig. 3f, g), showing that satellite cells lacking *Hira* are unable to replenish the muscle stem cell pool after muscle injury. Similar results were obtained 28 dpi in *Pax7CreErt2;Hirafl/fl* muscles when satellite cell number was normalized per area (Supplementary Fig. 3c, d), confirming the lack of capacity for the regenerating muscle to maintain the satellite cell pool. Consistently, the number of M-Cadherin-positive associated nuclei at 7 dpi, normalized to the unit area, was significantly decreased when *Hira* is deleted in satellite cells (Supplementary Fig. 3e). Consistent with the in vitro myogenic cell analysis, proliferation and cell death rates were not significantly affected in *Pax7CreErt2;Hirafl/fl* satellite cells (Supplementary Fig. 3f–h). Loss of satellite cells in *Pax7CreErt2;Hirafl/fl* injured muscles was associated with severely impaired regeneration capacity as shown by the high number of newly formed fibers 7 dpi labeled with embryonic myosin heavy chain (MYH3) antibody (Fig. 3h, i) and the failure to generate fibers of large cross-sectional area 28 dpi (Fig. 3j, k). These data demonstrate that muscle satellite cells lacking *Hira* fail to preserve the muscle stem cell pool and to efficiently regenerate damaged skeletal muscle.

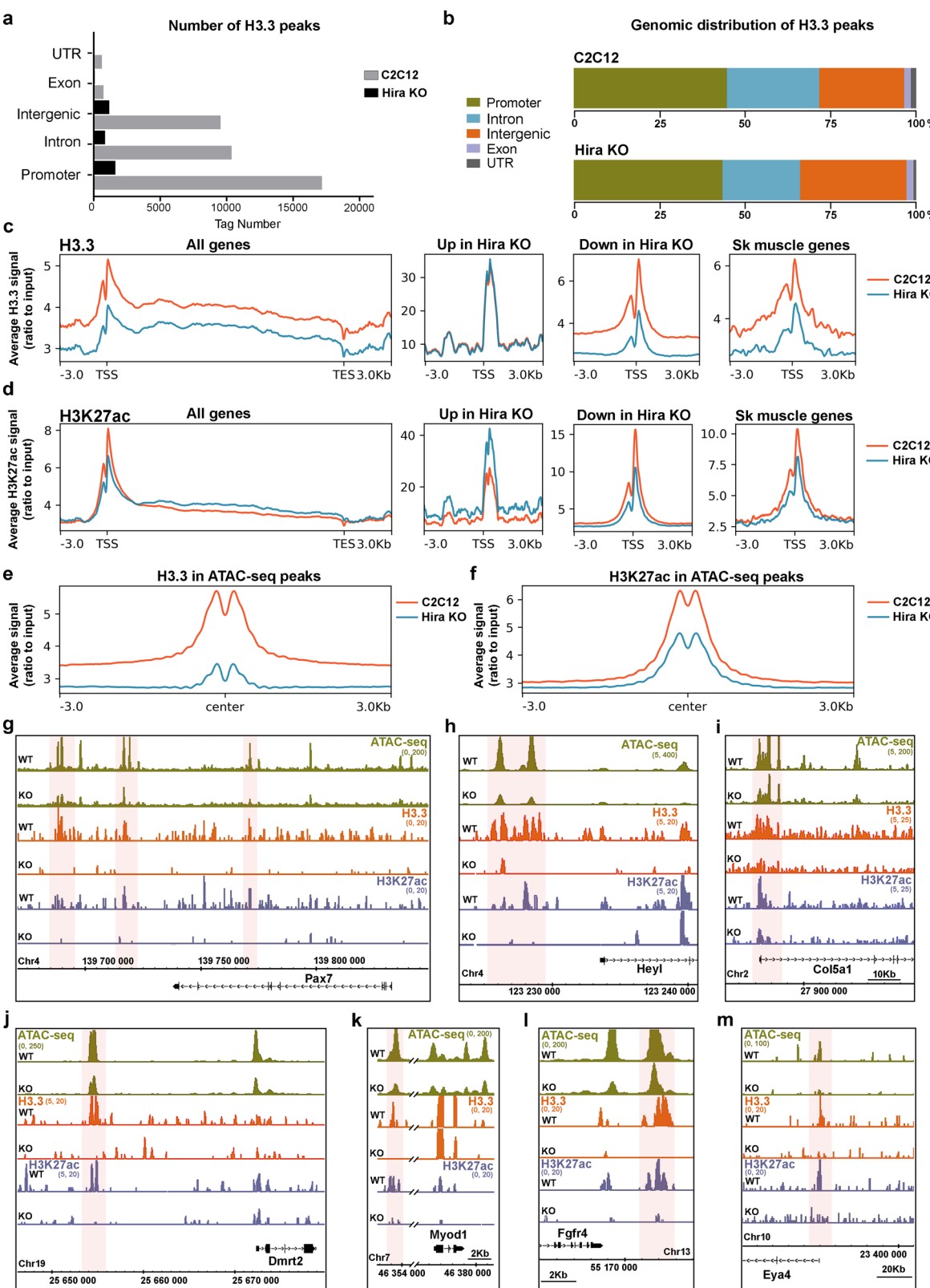

**Fig. 2 Loss of H3.3 in muscle gene regulatory regions in *Hira*-deficient cells. a** Total number of H3.3 called peaks (tag number) (*q*- value = 5e−2) in the ChIP-seq of C2C12 (gray, 38439 peaks) (*n* = 1) and *Hira*-KO cells (black, 3744 peaks) (*n* = 1). **b** Percentage of the total number of H3.3 peaks distributed in distinct genomic regions as indicated in C2C12 or *Hira*-KO cells. **c**, **d** ChIP-seq average signal profiles (ratio to input) in the promoter region (±3 kb around the TSS), TSS, gene body, and TES for H3.3 (**c**) and H3K27ac (**d**) in C2C12 (orange) and *Hira* KO (blue) shown for all genes, upregulated genes in *Hira* KO, downregulated in *Hira*-KO and skeletal muscle genes (GO:0007519). **e**, **f** ChIP-seq average signal profiles (ratio to input) of H3.3 (**e**) and H3K27ac (**f**) plotted on the control ATAC-seq peaks in C2C12 (orange) and *Hira* KO (blue). **g–m** ATAC-seq (green), ChIP-seq profiles for H3.3 (orange), and H3K27ac (gray) in the genomic loci of *Pax7* (**g**), *HeyL* (**h**), *Col5a1*(**i**), *Dmrt2* (**j**), *Myod1* (**k**), *Fgfr4* (**l**), and *Eya4* (**m**).

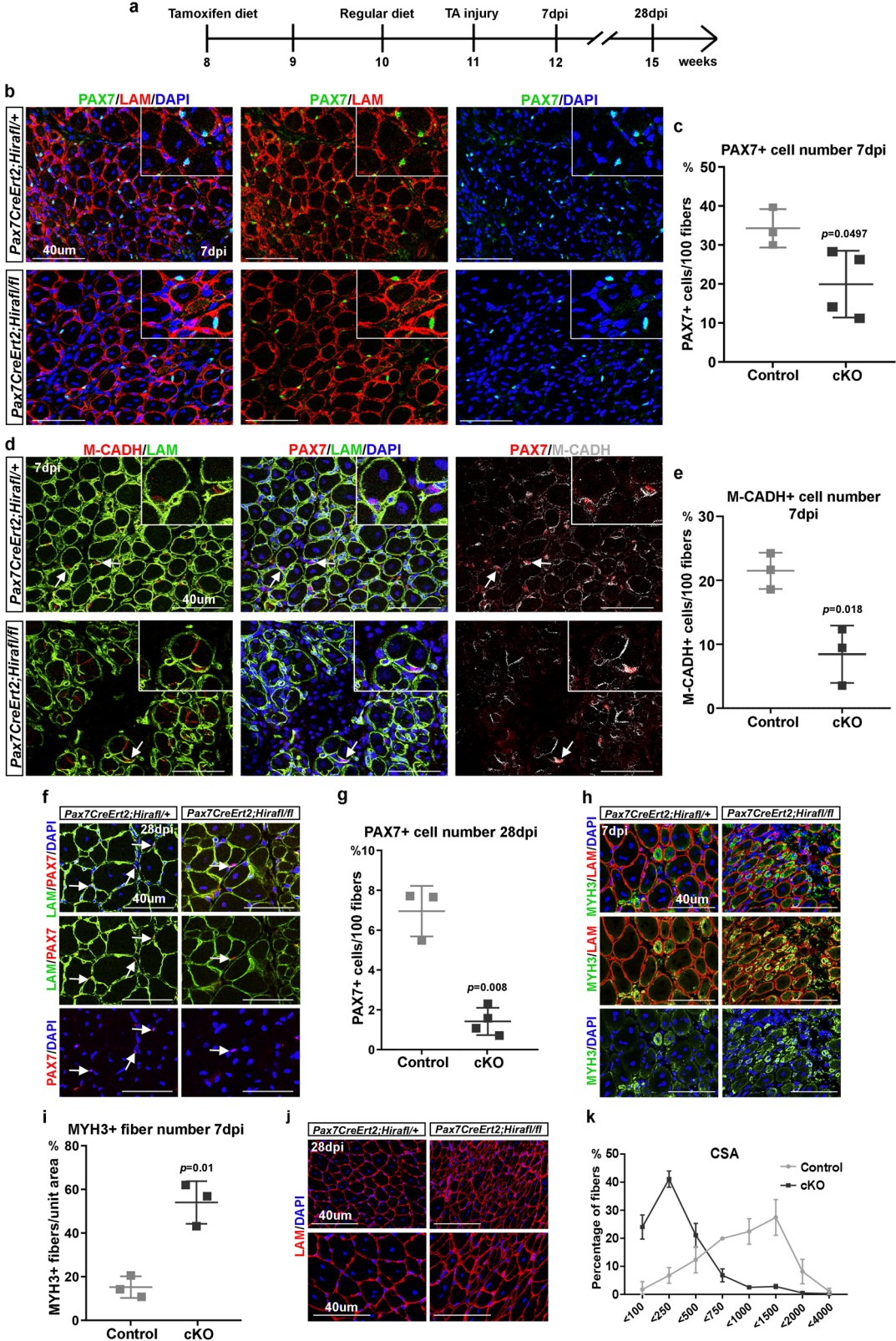

**Hira-deficient satellite cells lose myogenic cell identity**. Satellite cells lacking *Hira* are unable to sustain muscle regeneration. In order to analyze if *Hira*-deficient muscle stem cells display an alteration of lineage-specific gene expression, as observed in C2C12 cells, we performed RNA-seq on 5 dpi FACS-isolated satellite cells from control (*Pax7^CreErt2;Hira^fl/+*) and cKO (*Pax7^CreErt2;Hira^fl/fl*) mice. Sorted satellite cells were further

cultured for 48 h to homogenize the stem cell population and ensure activation of all cells. We observed expression changes in 2925 genes when comparing control and cKO samples (1297 downregulated and 1628 upregulated) (Fig. 4a, b). Gene ontology analysis of downregulated genes in satellite cells lacking *Hira* included terms related to skeletal muscle satellite cell differentiation, fiber development, and regeneration (Fig. 4c).

**Fig. 3** *Hira*-depleted satellite cells fail to regenerate injured muscle. **a** Schematic representation of experimental timeline. **b** Co-immunostaining using PAX7 (green), Laminin (LAM, red) and the nuclear marker DAPI (blue) in *Pax7^CreErt2;Hira^fl/+* and *Pax7^CreErt2;Hira^fl/fl* TA muscles 7 dpi. **c** Quantification of the PAX7-positive cell number per 100 fibers of **b** (control n = 3 mice, cKO n = 4 mice). Error bars, mean ± SD, two-sided unpaired *t*-test. **d** Co-immunostaining using PAX7 (red), M-Cadherin (M-CADH, red: first panel; gray: last panel), Laminin (LAM, green) and the nuclear marker DAPI (blue) in *Pax7^CreErt2;Hira^fl/+* and *Pax7^CreErt2;Hira^fl/fl* TA muscles 7 dpi. Arrows indicate M-CADH-positive cells. **e** Quantification of the M-CADH-positive cell number per 100 fibers in **d** (control n = 3 mice, cKO n = 3 mice). Error bars, mean ± SD, two-sided unpaired *t*-test. **f** Co-immunostaining using PAX7 (red), Laminin (LAM, green), and the nuclear marker DAPI (blue) in *Pax7^CreErt2;Hira^fl/+* and *Pax7^CreErt2;Hira^fl/fl* TA muscles 28 dpi. Arrows indicate PAX7-positive cells. **g** Quantification of the PAX7-positive cell number per 100 fibers in **f** (control n = 3 mice, cKO n = 4 mice). Error bars, mean ± SD, two-tailed unpaired *t*-test. **h** Co-immunostaining using MYH3 (green), Laminin (LAM, red), and the nuclear marker DAPI (blue) in *Pax7^CreErt2;Hira^fl/+* and *Pax7^CreErt2;Hira^fl/fl* TA muscles 7 dpi. **i** Quantification of the MYH3-positive fibers per area in **h** (control n = 3 mice, cKO n = 3 mice). Error bars, mean ± SD, two-tailed unpaired *t*-test. **j** Immunostaining using Laminin (LAM, red) and the nuclear marker DAPI (blue) in *Pax7^CreErt2;Hira^fl/+* and *Pax7^CreErt2;Hira^fl/fl* TA muscles 28 dpi. **k** Quantification of the CSA in **j** as percentage of total fibers (control n = 3 mice, cKO n = 3 mice). Error bars, mean ± SD. Scale bars, 40 µm.

In contrast, gene ontology of upregulated genes was linked to angiogenic, cardiac, and endothelial processes, which suggests an increase in the expression of alternative lineage genes in the absence of HIRA in satellite cells (Fig. 4d). These results are consistent with the C2C12 gene ontology analysis (Fig. 1i, j). In the *Hira*-deficient satellite cells, muscle stem cell markers and myogenic markers are consistently downregulated (Fig. 4e, f), while genes from cardiac, endothelial, and other alternative lineages are upregulated (Fig. 4e, f). Moreover, satellite cells lacking *Hira* display positive VE-Cadherin (*Cdh5* gene) immunostaining (Supplementary Fig. 3i, j). These results suggest that a similar mechanism is operating in C2C12 and in satellite cells to sustain the myogenic identify of these cells. In order to test this, we performed H3.3 ChIP-RT-qPCR in satellite cells and analyzed the known regulatory regions of *Myod1* (−20 kb) and *Fgfr4* (+19.2 kb)[19,20], the genomic regions of *Pax7* (+62.4 kb) and *Eya4* (−380 bp) that display H3.3 enrichment in C2C12 cells (Fig. 2g, m) and the negative control for H3.3 binding (*Myod1*, −15 kb)[11]. All of these myogenic genes are downregulated in satellite cells (Fig. 4e, f). We observed that similar to C2C12 cells, H3.3 recruitment to the regulatory regions of these genes is lost in *Hira*-deficient satellite cells (Fig. 4g). Taken together, we conclude that H3.3 deposition by HIRA is required to maintain muscle stem cell gene expression and sustain a myogenic identity in satellite cells.

**Alternative lineage genes are expressed in *Hira*-KO myoblasts.** We identified a significant number of non-myogenic genes that were upregulated in *Hira*-deficient myogenic cells and that were previously shown to be silent in muscle cells, displaying either no histone marks or a bivalent state in C2C12[15]. Bivalent promoters are associated with silent genes characterized by both active H3K4me3 and repressive H3K27me3 histone marks, remaining in a poised state that can be resolved into an active or repressed state upon differentiation[25]. To define the epigenetic status of the genes upregulated in *Hira*-deficient cells, we performed ChIP-seq for H3K4me3 and H3K27me3 in control and *Hira*-mutant C2C12 myogenic cells. We observed that H3K4me3 signal intensity was strongly increased in promoters, TSS, and gene bodies of all protein-coding genes in *Hira*-deficient myogenic cells, while H3K27me3 signal was unchanged (Supplementary Fig. 4a, b). In addition, we analyzed the bivalency-associated marks in the promoter regions of upregulated genes, identifying a disruption of the H3K4me3 and H3K27me3 signal balance consistent with their induced expression (Fig. 5a). In contrast, the promoters of the downregulated genes do not show a bivalent profile in C2C12 or in *Hira*-KO cells (Fig. 5a). The Hoxb cluster, which was strongly upregulated in *Hira*-deficient myogenic cells (Fig. 1k, l), was previously described as presenting only repressive marks in C2C12 cells;[15] on the contrary, we observed bivalent features at these loci with H3K27me3 and H3K4me3 marks

(Fig. 5b), similarly to quiescent satellite cells[26]. Upon *Hira* deletion, the Hoxb cluster showed a concomitant increase in H3K4me3 modification and loss of H3K27me3 mark (Fig. 5b). Other bivalent genes, associated with adipogenic fate (*Sox6*, *Cebpa*) had increased H3K4me3 signal in their promoters but did not display modifications in H3K27me3 upon *Hira* deletion, indicating, together with the general absence of H3K27me3 changes in all genes, that the major histone mark associated with activation and expression of the bivalent genes is H3K4me3 (Supplementary Fig. 5c, d). Upregulated genes in *Hira*-deficient cells belonging to neuronal (*Nefl*, *Nefm*), endothelial (*Cdh5*, *Angpt2*), mesenchymal (*Pdgfra*), or osteogenic lineages (*Runx2*) did not display histone marks in myogenic cells (Fig. 5c–f and Supplementary Fig. 4e, f), associated with an absence of RNA expression of these genes. However, upon *Hira* deletion, these genes acquired H3K4me3 marks and became expressed (Fig. 5c–f and Supplementary Fig. 4e, f). Since MLL1/MLL2 methyl-transferase complex is responsible for H3K4 trimethylation[27], we investigated if this complex mediated the activation of alternative lineage genes in myogenic cells depleted of *Hira*. We analyzed the expression of *Mll1* (*Kmt2a*) and *Mll2* (*Kmt2b*) genes in *Hira*-deficient cells and found that they were upregulated 1.9- and 1.5-fold, respectively, compared with controls (Fig. 5g, h). To investigate if the upregulation of *Mll1/Mll2* genes is promoting the expression of alternative lineage genes when *Hira* is deleted in myogenic cells, we knocked out either *Mll1* or *Mll2* in *Hira*-deficient cells (Supplementary Fig. 4g, h). Deleting *Mll1* or *Mll2* significantly decreased their RNA levels and both *Hira/Mll1* dKO and *Hira/Mll2* dKO lacked MLL1 protein (Supplementary Fig. 4g, h). Strikingly, abolishing *Mll1* or *Mll2* expression in *Hira* mutant cells prevented the expression of non-myogenic lineage genes (*Cdh5*, *Nefl*, *Nefm*) and Hoxb cluster gene *Hoxb13* (Fig. 5i). The expression levels of *Pax7*, shown to be regulated by MLL1 in activated satellite cells[28] were partially rescued in *Hira/Mll1* and *Hira/Mll2* dKO cell lines compared to *Hira* KO alone (Supplementary Fig. 4i). To evaluate if the expression of non-myogenic genes was linked to the H3K4 trimethyltransferase activity of MLL1/MLL2, we performed H3K4me3 ChIP-RT-qPCR and analyzed the promoter regions of *Cdh5*, *Nefl*, and *Hoxb13*. We confirmed that at these loci the enrichment of the H3K4me3 histone mark was reduced in *Hira/Mll1* dKO cells compared to *Hira* mutant cells (Fig. 5j), confirming that increased H3K4me3 is linked to MLL1/MLL2 expression. Since H3K4me3 histone modification is increased across the genome in *Hira*-KO cells both in genes that are upregulated and downregulated (Supplementary Fig. 4a), we asked whether other histone modifications could be associated with the expression of alternative lineage genes. Since loss of H3.3 and decreased H3K27ac modification in myogenic gene loci is associated to decreased expression in *Hira*-KO cells, we asked whether acetylation in promoter regions of alternative lineage genes

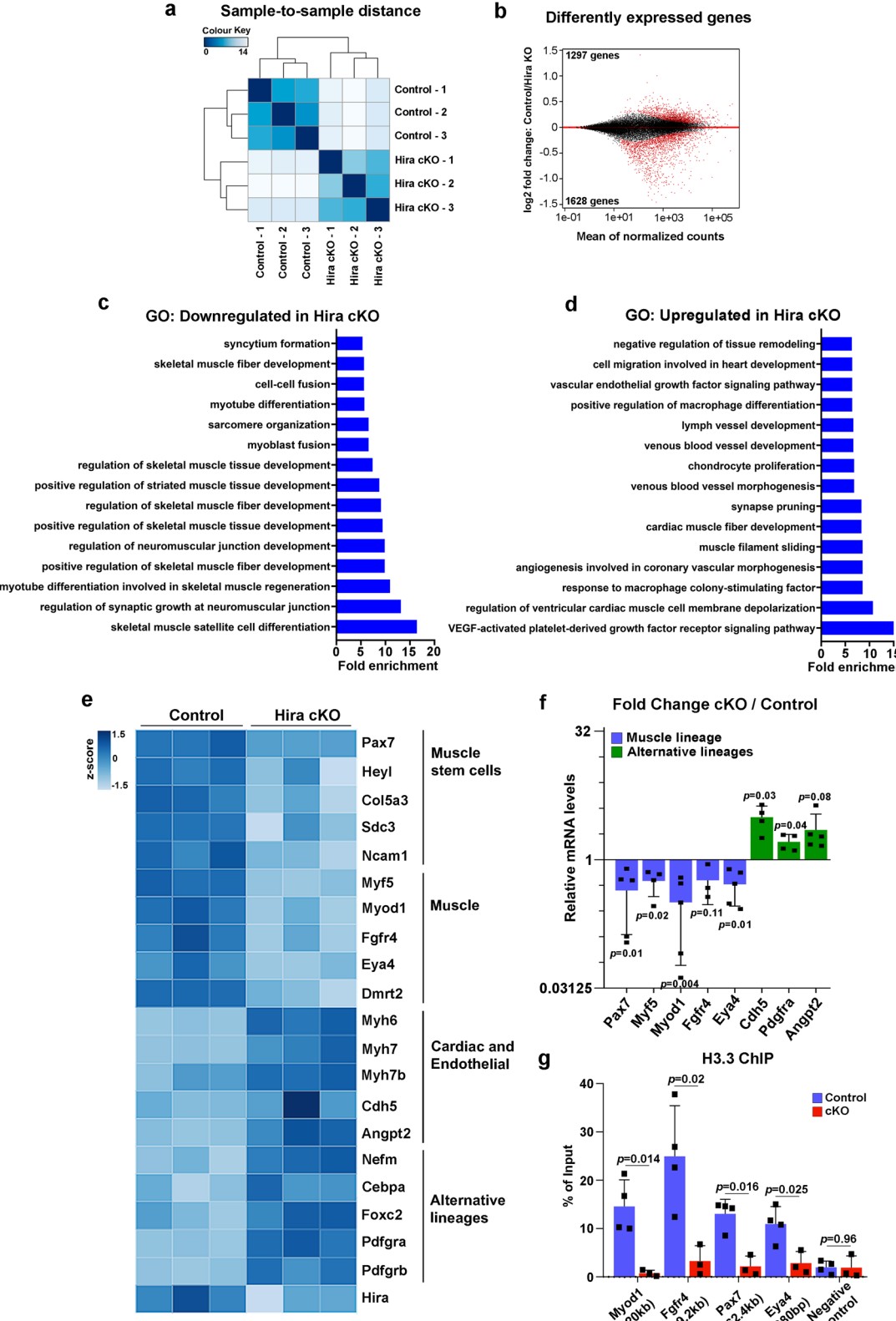

**Fig. 4 *Hira*-deficient satellite cells lose myogenic identity. a** Differential expression analysis of the RNA-seq analysis performed in 5 dpi satellite cells (*n* = 3 independent RNA samples for control and cKO). **b** MA-plot of control over *Hira* cKO satellite cells RNA-seq data. Significantly dysregulated genes are highlighted in red (FDR < 0.05). **c**, **d** Gene ontology analysis for biological processes of the downregulated (**c**) and upregulated (**d**) genes in cKO. Selected enriched terms are presented according to the fold enrichment. **e** Heatmap with the normalized reads per gene of control and *Hira* cKO RNA-seq data in individual triplicates. **f** RT-qPCR analyses in satellite cells (5 dpi) from *Pax7^{CreErt2}*;*Hira^{fl/+}* and *Pax7^{CreErt2}*;*Hira^{fl/fl}* (*n* = 5 mice for Pax7, Myod1, Eya4, Angtp2; *n* = 4 mice for Myf5, Cdh5, Pdgfra; *n* = 3 mice for Fgfr4). For each gene, the mRNA levels of the control cells were normalized to 1. Error bars, mean ± SD, two-tailed paired *t*-test. **g** ChIP-RT-qPCR for H3.3 in *Myod1* (−20 kb), *Fgfr4* (+19.2 kb), *Pax7* (+62.4 kb), *Eya4* (−380 bp) and the negative control (−15 kb *Myod1*) of control (blue, *n* = 4) and *Hira*-depleted (red, *n* = 3) satellite cells (5 dpi). Error bars, mean ± SD, two-tailed unpaired *t*-test.

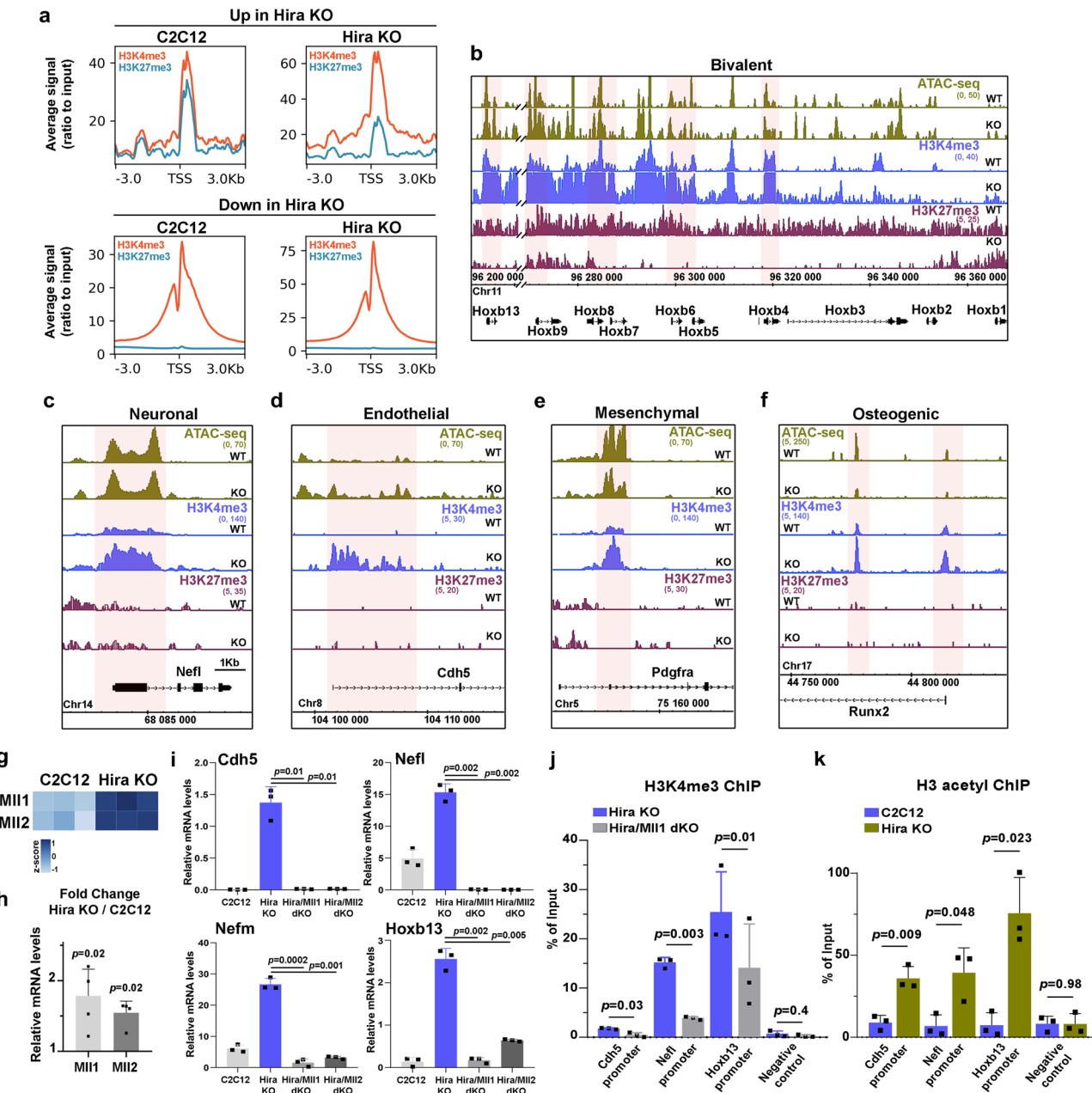

**Fig. 5 Activation of alternative lineage genes in *Hira*-KO cells is dependent on MLL1/MLL2 complex. a** ChIP-seq average signal profiles (ratio to input) in the promoter region (±3 kb around the TSS) and TSS for H3K4me3 and H3K27me3 in C2C12 and *Hira* KO, plotted with the upregulated or the downregulated genes (*n* = 1). **b**–**f** ATAC-seq (green) and ChIP-seq profiles for H3K4me3 (blue) and H3K27me3 (dark red) in the Hoxb cluster (**b**) and in the genomic loci of *Nefl* (**c**), *Cdh5* (**d**), *Pdgfra* (**e**), and *Runx2* (**f**) genes. Gene lineages are indicated. **g** Heatmap with the normalized reads per gene of C2C12 and *Hira*-KO RNA-seq data for *Mll1* and *Mll2*. **h** RT-qPCR analyses of the mRNA expression levels of *Mll1* (light gray) and *Mll2* (dark gray) in C2C12 (*n* = 4 independent RNA samples) and *Hira* KO (*n* = 4 independent RNA samples) cells. For each gene, the mRNA levels of the control cells were normalized to 1. Error bars, mean ± SD. two-tailed paired *t*-test. **i** RT-qPCR analyses of the mRNA expression levels of *Cdh5*, *Nefl*, *Nefm*, and *Hoxb13*, in C2C12, *Hira* KO, *Hira/Mll1* dKO, and *Hira/Mll2* dKO cells (*n* = 3 independent RNA samples per condition). Error bars, mean ± SD, two-tailed unpaired *t*-test. **j** ChIP-RT-qPCR for the H3K4me3 histone mark on the promoter region of *Cdh5*, *Nefl*, and *Hoxb13* genes and in the negative control (intronic region of *Cdh5*) of *Hira* KO (blue) and *Hira/Mll1* dKO (gray) cell lines (*n* = 3 independent biological samples per condition). Error bars, mean ± SD, two-tailed paired *t*-test. **k** ChIP-RT-qPCR for the H3ac (pan-acetyl) histone mark on the promoter region of *Cdh5*, *Nefl*, and *Hoxb13* genes and in the negative control (intronic region of *Cdh5*) of control C2C12 (blue) and *Hira* KO (green) (*n* = 3 independent biological samples per condition). Error bars, mean ± SD, two-tailed unpaired *t*-test.

could be associated with their activation. Thus, we performed H3-acetyl ChIP-RT-qPCR (that recognizes acetylation in all N-terminal residues of H3 histones) at the promoter regions of *Cdh5*, *Nefl*, and *Hoxb13*. We observed that in *Hira* mutant cells, promoters of alternative lineage genes and Hox genes display an

increased acetylation compared with control cells (Fig. 5k). Taken together these data demonstrate that, HIRA is required to prevent the expression of silent alternative non-myogenic lineage genes by (1) inhibiting *Mll1/Mll2* expression and consequently preventing the H3K4 methyltransferase activity of

these proteins and by (2) preventing acetylation of H3 histones at promoter regions of those genes.

## Discussion

In the murine myogenic cell line C2C12, it was previously shown that the histone chaperone activity of HIRA is required to deposit H3.3 in the core enhancer region of *Myod1*, to sustain its expression during differentiation[11]. In this myoblast cell line, forced expression of the canonical H3 variant H3.1 disrupts the balance of the H3.3/H3.1 in the nucleosomes, which inhibits myogenic gene expression and suppresses the differentiation potential of the cells[10]. Here we demonstrate that HIRA plays a considerably broader function by providing an appropriate chromatin landscape that promotes muscle gene expression via H3.3 incorporation, which is required to maintain the active transcription-associated mark H3K37ac at myogenic loci.

Whether H3.3 deposition-associated regulation of myogenic gene expression is an intrinsic function of the histone chaperone HIRA or if other factors are involved remains to be determined. The ubiquitous chromodomain helicases CHD1 and CHD2, which are recruited to active chromatin regions associated with TSS and enhancers in a transcription-coupled manner participate in the deposition of H3.3[9,29,30]. Moreover, CHD2 deletion leads to reduced H3.3 occupancy at TSS and enhancers in a human lymphoblast cell line[29]. It was previously shown that MYOD physically interacts with CHD2 and that this interaction is required for H3.3 incorporation in the regulatory regions of myogenic terminal differentiation genes, such as MYOG, regulating their expression[9]. Although CHD1 physically associates with HIRA to mediate H3.3 deposition into chromatin in human lymphoblast cell lines, CHD1 is not involved in H3.3 deposition in *Myod1* promoter or upstream regulatory regions in myogenic cells[11,30]. The H3.3 histone chaperone DAXX regulates the deposition of this histone variant in constitutive heterochromatin but it was also shown to deposit H3.3 in regulatory regions of neuronal genes during development, regulating their expression[7,8,12]. However, the role of DAXX as a histone chaperone of H3.3 during myogenesis remains to be evaluated. Altogether, HIRA appears to be the main histone chaperone required for H3.3 deposition in the myogenic lineage, while CHD2 could act either as a co-factor or incorporate H3.3 at the locus of MYOD downstream target genes.

The epigenetic determinants of lineage-committed adult stem cells are poorly understood compared to those of ESCs. In this study, we identify two major epigenetic differences between the adult myogenic stem cell lineage and what has been identified for ESCs: (1) the drastic transcriptomic changes when HIRA is deleted vs. no changes in ESCs[8,14] and (2) the general loss of H3.3 deposition in the absence of HIRA in myoblasts vs. loss of H3.3 deposition in specific genomic loci in ESCs[8]. Although the pluripotent transcriptomic state of ESCs can be maintained in the absence of HIRA and depletion of H3.3 in TSS and gene bodies[8], the deposition of H3.3 in regulatory regions of developmental genes is required for ESCs differentiation into the different germ layers[14,31]. In addition, while the deposition of H3.3 in promoter-distal enhancers (±3 kb around the TSS) of ESCs is required for H3K27 acetylation and enhancer activation, an effect in gene expression is only observed when the cells are triggered to differentiate[14]. These results suggest that H3.3 plays a major role in lineage commitment and maintenance, consistent with our observation that H3.3 is required to preserve myogenic gene expression and lineage potential of skeletal muscle stem cells. Although, it remains to be investigated if this mechanism is specific to the myogenic lineage or also operates in other tissues.

In this study, we combined the analysis of skeletal muscle stem cells with a myogenic cell line (C2C12) to address the requirement of HIRA-H3.3 in the acquisition of myogenic commitment and in its maintenance, respectively. We showed that the deposition of H3.3 by HIRA in myogenic genomic loci is required for gene expression in activated satellite cells upon muscle injury, as well as for the maintenance of muscle identity in already committed cells (C2C12). In vivo deletion of *Hira* reduces satellite cell numbers and severely compromises muscle regeneration. We showed that this is associated with loss of myogenic gene expression and muscle stem cell markers in the absence of HIRA. The percentage of proliferating PAX7-positive cells is not altered in *Hira*-deficient satellite cells, yet we cannot exclude that loss of muscle stem cells could be linked to an unbalanced asymmetric cell division, which is known to be required for replenishing the quiescent pool of satellite cells[32,33]. Given the consistency between the in vivo and in vitro data, we believe that the regeneration impairment phenotype is associated with the transcriptomic changes observed in satellite cells-lacking *Hira* and therefore to the role of HIRA in the maintenance of myogenic identity. Although, an alternative hypothesis in the context of muscle regeneration could be linked to the production of impaired myofibers. Future studies will evaluate the impact of myofiber specific conditional ablation of *Hira* to clarify this point.

We observed a striking and specific loss in muscle stem cell lineage gene expression linked to the decreased H3.3 deposition in *Hira*-deficient cells. Yet, the activation of gene expression from non-muscle lineages in cells depleted for HIRA did not occur through changes in H3.3 deposition at their loci and was associated with the increased and global activity of the H3K4 methyltransferases MLL1 and MLL2. Indeed, the MLL1/MLL2 complex has been shown to play a major role in lineage commitment during development, associated with regulation of Hoxb cluster expression, demonstrating that this methyltransferase complex is associated with the specification of new lineages[34–36].

In this study, we identified an epigenetic mechanism by which HIRA, via H3.3 incorporation, stabilizes the myogenic identity of muscle stem cells and committed myoblasts by maintaining their gene expression active and by silencing alternative lineage genes through MLL1/MLL2 inhibition. Our study therefore supports a crucial role for HIRA-mediated H3.3 incorporation that subsequently stabilizes the active transcription-associated histone mark H3K27ac in lineage-specific cells to maintain the balanced expression of tissue-specific genes.

## Methods

**Mouse lines**. All mice were kept under pathogen-free conditions with a 12 h/12 h light/dark cycle and a constant ambient temperature (22 °C) and humidity (34%). Adult male mice aged between 8 and 12 weeks were used as heterozygous *Pax7^CreErt2;Hira^fl/+* (controls) or homozygous *Pax7^CreErt2;Hira^fl/fl* for the *Hira* floxed allele[22,23]. Animals were handled according to the European Community guidelines, implementing the 3Rs rule. Protocols were validated by the ethic committee of the French Ministry, under the reference number APAFIS#13695-2018021408521124.v2.

**Tamoxifen administration and muscle injury**. Mice were put on a solid tamoxifen diet (400 mg/kg of food) for 2 weeks and then kept 1 week with regular food. Mice were anesthetized intra-peritoneally with 3.5 µL/g of Ketamine/Xylazine at 20 and 10 mg/mL, respectively. The TA muscles were injected with 50 µl of 0.6% BaCl$_2$ (Sigma, 202738).

**Immunohistochemistry**. Cryosections of 10 µm were performed on TA muscles previously dissected and frozen in liquid-nitrogen-cooled isopentane 7 or 28 dpi. The muscle sections or the cultured cells were fixed in 4% PFA for 15 min at room temperature, permeabilized in cold methanol for 6 min, boiled in citrate buffer (Dako, S1699) for epitope retrieval and blocked with 5% immunoglobin G (IgG)-free BSA (Jackson, 001-000-162) for 1 h at room temperature. To reduce background, the slides were incubated 30 min at room temperature with anti-mouse IgG Fab fragment (Jackson, 115-007-003). The sections were incubated with

primary antibodies (Supplementary Table 1) diluted in 5% IgG-free BSA overnight at 4 °C. Secondary antibody incubation (Invitrogen, Alexa-Fluor) was performed 1 h at room temperature followed by DAPI (Sigma, D9542) staining (1:5000) 10 min at room temperature. EdU staining reaction was performed according to manufacturer's guidelines (Invitrogen, EdU Click-iT PLUS Kit, C10640).

**Western blot**. Cells were lysed in RIPA buffer (50 mM Tris HCl (Sigma, T5941), 150 mM NaCl (Sigma, 9625), 1% Igepal (Sigma, I8896), 0.5% Sodium deoxycholate (Sigma, D6750), 0.1% SDS (Sigma, L3771), 1 mM EDTA (Sigma, E5134), 1x Protease Inhibitor Cocktail (Roche, 04-693-116-001); pH8) for protein extraction. Overall, 10 μg of protein were used per sample. The blocking of the membrane was performed in 0.5% of gelatin from cold water fish (Sigma, G7765) and 5% Tween20 (Sigma, P4416) in PBS (Invitrogen, 003002) 1 h at room temperature. Primary antibodies were diluted in blocking solution and incubated 2 h at room temperature (Supplementary Table 1). Secondary antibodies were diluted in blocking solution and incubated 1 h at room temperature (Supplementary Table 1). Revelation reaction was performed 5 min at room temperature using SuperSignal West Pico (ThermoScientific, 35060). Western blot quantification was performed using Image J[37]. Uncropped and unprocessed blots are displayed in the Source Data file.

**Muscle dissection and satellite cell isolation**. FACS-isolation of satellite cells was performed as following[18]. Muscles were dissected, minced, and incubated in digestion buffer: HBSS (Gibco, 14025092), 0.2% BSA (Sigma, A7906), 2 μg/ml Collagenase A (Roche, 11088793001), and 3.25 μg/ml Dispase II (Roche, 04942078001) for 2 h at 37 °C, and purified by filtration using 100 μm and 40 μm cell strainers (BD Falcon, 352360 and 352340). Cell suspensions were incubated 30 min on ice with the following antibodies: Alexa700-anti-ITGA7, BV421-anti-CD34, PE-Cy7-anti-TER119, PE-Cy7-anti-CD45, and PE-anti-Ly-6/E (SCA1) (Supplementary Table 1). Cells were sorted by gating CD34 + ITGA7+ double positive lineage as shown in Supplementary Fig. 5.

**Cell culture of C2C12 cells**. C2C12 cell line (ATCC: CRL1772), obtained from DSMZ Germany, were grown in DMEM (Gibco, 41966029) supplemented with 10% fetal calf serum (FCS) (Eurobio, CVFSVF00-01) and 1% Pen/Strep (Gibco, 15070063). For differentiation assays, C2C12 were seeded at a confluence of 80% with 2% FCS for 3 days. For EdU proliferation analysis, cells were incubated with 10 mM EdU (Invitrogen, A10044) for 24 h (Invitrogen, EdU Click-iT PLUS Kit, C10640).

**Cell culture of satellite cells**. FACS-isolated satellite cells were plated on Matrigel (Corning, 354248)-coated dishes and cultured in DMEM (Gibco, 41966029) with 20% FCS (Eurobio, CVFSVF00-01), 1% Pen/Step (Gibco, 15070063), and 4 ng/mL basic FGF (bFGF) (Peprotech, 450-33).

**Generation of KO C2C12 line using CRISPR-Cas9**. *Hira*, *Mll1*, and *Mll2* knockouts were generated by CRISPR/Cas9 using the pU6-(BbsI)_CBh-Cas9-T2A-mCherry plasmid, a gift from Ralf Kuehn (Addgene plasmid # 64324; http://n2t. net/addgene:64324; RRID:Addgene_64324)[38]. The single guide RNAs (sgRNAs) were obtained from the sgRNA optimized library[39]. The primers were designed with added BbsI (ThermoScientific, FD1014) restriction sites and an extra G/C for increased hU6 promoter efficiency (Supplementary Table 2), annealed and ligated to BbsI-linearized pU6-CBh-Cas9-T2A-mCherry plasmid using the T4-ligase (ThermoScientific, EL0011). A total of $1.5 \times 10^5$ C2C12 cells were transfected with 7 μg of plasmid with Lipofectamine LTX PLUS reagent (Invitrogen, 15338030) and 3 days post transfection the cells were FACS-isolated for mCherry-positive. The sorted cells were seeded at a low confluence (500 cells per 10 cm dish) in dishes previously coated with 0.1% gelatin from pork skin (Sigma, G1890). Individual clones were isolated, expanded, genotyped by PCR and sequenced to confirm the mutation.

**RNA extraction, cDNA synthesis and RT-qPCR**. A minimum of $2 \times 10^5$ C2C12 cells, cultured in high serum-containing medium and at low density, were collected per sample for RNA extraction (Macherey Nagel, 740955) following the manufacturer protocol. A minimum of $10^5$ FACS-isolated satellite cells were collected per sample for RNA extraction. Reverse transcription was performed using the SuperScript III Reverse Transcriptase (Invitrogen, 18080-093) following the manufacturer's guidelines. RT-qPCR was performed using the PowerUp SYBR Green Master Mix (Applied Biosystems, A25742). The relative mRNA levels were calculated using the $2^{\wedge-\Delta\Delta Ct}$ method[40]. The ΔCt were obtained from Ct normalized to the housekeeping gene *Tbp* levels in each sample. The RT-qPCR primers used are listed in the Supplementary Table 2.

**RNA-sequencing of C2C12 cells**. RNA was prepared as described for RNA extraction and sent to Integragen. Libraries were prepared with TruSeq Stranded Total RNA Sample preparation kit according to supplier recommendations. Briefly, the key stages of this protocol are successively, the removal of ribosomal RNA fraction from 1 μg of total RNA using the Ribo-Zero Gold Kit; fragmentation using divalent cations under elevated temperature to obtain `300 bp pieces; double strand

cDNA synthesis using reverse transcriptase and random primers, and finally Illumina adapters ligation and cDNA library amplification by PCR for sequencing. Sequencing was carried out on paired-end 75 bp of Illumina HiSeq4000.

**RNA-sequencing of satellite cells**. RNA was prepared as described for RNA extraction and sent to the IMRB (Institut Mondor de Recherche Biomédicale) genomic platform. Libraries were prepared with TruSeq Stranded Total Library preparation kit according to supplier recommendations. Briefly, the key stages of this protocol are successively, the removal of ribosomal RNA fraction from 400 ng of total RNA using the Ribo-Zero Gold Kit; fragmentation using divalent cations under elevated temperature to obtain ~300 bp pieces; double strand cDNA synthesis using reverse transcriptase and random primers, and finally Illumina adapters ligation and cDNA library amplification by PCR for sequencing. Sequencing was carried out on single-end 75 bp of Illumina NextSeq500.

**RNA-sequencing analysis**. The RNA-seq analysis was performed using the Galaxy web platform[41], public server https://usegalaxy.org. The FASTQ files were uploaded in Galaxy and formatted as Sanger using the FASTQ groomer tool[42]. Quality of the data was analyzed with FastQC tool v0.72. FASTQ Sanger files were aligned to the mm10 mouse genome using the built-in index of BOWTIE2 v2.3.4.2[43]. Genes were counted using featureCounts v1.6.3 + galaxy2[44] and differently expressed genes were determined by DESeq2 v2.11.40.6 + galaxy1[45], obtaining also the MA-Plot and the sample-to-sample distances plot. Gene ontology analyses were performed on http://geneontology.org/ using the biological process option. The gene expression heatmap was created with displayR software with normalized reads for each triplicate.

**Chromatin immunoprecipitation (ChIP)**. Native ChIP was performed as following[46]. Briefly, C2C12 cells, cultured in high serum-containing medium and at low density, or FACS-isolated satellite cells (as described above) were trypsinized, washed, and subjected to nuclei isolation, chromatin fragmentation using Micrococcal Nuclease (MNase) (Sigma, N5386) and nucleosome purification by Hydroxyapatite (BioRad, 158-2000) chromatography. Immunoprecipitation was performed overnight at 4 °C with 6 μg of chromatin and 5 μg of antibody (Supplementary Table 1), previously incubated with Protein-A- (for H3K27ac, H3K4me3, H3K27me3, and H3ac antibodies) or Protein-G-coated (for H3.1 and H3.3 antibodies) magnetic beads (Diagenode, C03010020, and C03010021, respectively) and analyzed by RT-qPCR as percentage of the input or by sequencing. For sequencing, samples were sent to the IRCM (Institut de recherches cliniques de Montréal) molecular biology platform. Library was prepared using KAPA Hyper Prep Kits with PCR Library Amplification/Illumina series (Roche, 07962363001) with IDT for Illumina TruSeq DNA-RNA UD 96 Indexes (UDI) (Illumina, 2023784) and quantified by RT-qPCR using NEBNext Library Quant Kit for Illumina (NEB, E7640AA). Sequencing was carried out on paired-end 50 bp of Illumina HiSeq4000.

**ChIP-sequencing analysis**. The ChIP-seq analysis was performed using the Galaxy web platform[41] public server https://usegalaxy.org. The FASTQ files were uploaded in Galaxy and formatted as Sanger using the FASTQ groomer tool[42]. Quality of ChIP-seq data was analyzed with FastQC tool v0.72. FASTQ Sanger files were aligned to the mm10 mouse genome using the built-in index of BOWTIE2 v2.3.4.2[43]. The peaks were called with MACS2 v2.1.1.2[47] using paired-end BAM files with the cutoff *q* value 5−e2 and the broad region calling off, the input was used as control. MACS2 resulting bedgraph file was converted to bigwig format using the wig/bedgraph-to-bigwig converter v1.1.1 for visualization in the Integrated Genome Browser v9.1.4[48]. The called peaks were annotated using the ChIPseeker v1.18.0 tool[49]. Average signal graphs were performed using computeMatrix Galaxy version 3.3.2.0.0[50] to prepare data for plotting using plotProfile Galaxy version 3.3.2.0.0[50].

**ATAC-sequencing**. C2C12 cells, cultured in high serum-containing medium and at low density, were harvested and frozen in culture media containing 10% FCS (Eurobio, CVFSVF00-01) and 10% DMSO (ThermoScientific, 20688). Cryopreserved cells were sent to Active Motif to perform the ATAC-seq assay. The cells were then thawed in a 37 °C water bath, pelleted, washed with cold PBS, and tagmented as previously described[51], with some modifications based on[52]. Briefly, cell pellets were resuspended in lysis buffer, pelleted and tagmented using the enzyme and buffer provided in the Nextera Library Prep Kit (Illumina). Tagmented DNA was then purified using the MinElute PCR purification kit (Qiagen), amplified with ten cycles of PCR and purified using Agencourt AMPure SPRI beads (Beckman Coulter). Resulting material was quantified using the KAPA Library Quantification Kit for Illumina platforms (KAPA Biosystems) and sequenced with PE42 sequencing on the NextSeq 500 sequencer (Illumina).

**ATAC-sequencing analysis**. Reads were aligned using the BWA algorithm (mem mode; default settings). Duplicate reads were removed, only reads mapping as matched pairs and only uniquely mapped reads (mapping quality ≥ 1) were used for further analysis. Alignments were extended in silico at their 3′-ends to a length

of 200 bp and assigned to 32-nt bins along the genome. The resulting histograms (genomic "signal maps") were stored in bigWig files. Peaks were identified using the MACS 2.1.0 algorithm at a cutoff of $p$ value 1e−7, without control file and with the –nomodel option. Signal maps and peak locations were used as input data to Active Motif's proprietary analysis program, which creates Excel tables containing detailed information on sample comparison, peak metrics, peak locations, and gene annotations.

**Image capturing, quantification, and statistical analysis**. Image capturing was performed using a Zeiss LSM 800 confocal microscope with the associated Zeiss Zen Lite v2.3software. Quantifications were performed in at least five pictures taken randomly within each sample, per experiment, using Image J[37] and statistics were done with GraphPad Prism version 8 and Excel. The statistical test performed in each analysis is described in the associated figure legend.

**Reporting summary**. Further information on research design is available in the Nature Research Reporting Summary linked to this article.

## Data availability

The RNA-seq, ChIP-seq, and ATAC-seq sequencing data that support the findings of this study have been deposited in GEO NCBI with the accession codes "GSE161056" and "GSE167911". All other relevant data supporting the key findings of this study are available within the article and its Supplementary Information files or from the corresponding author upon reasonable request. Source data are provided with this paper. A reporting summary for this article is available as a Supplementary Information file. Source data are provided with this paper.

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

## Acknowledgements

We thank Matthew Borok, Despoina Mademtzoglou, Valentina Taglietti, Philippos Mourikis, and Delphine Duprez for reading and commenting on the manuscript. We thank Carole Conejero and Stéphane Kerbrat from the IMRB genomic platform. We thank Adeline Henry, Aurélie Guguin, and Odile Ruckebusch from the IMRB cytometry platform. We thank Odile Neyret from the IRCM molecular biology platform. We thank the animal facilities EP3 from IMRB and TAAM from CDTA. This work was supported by funding to FR from Association Française contre les Myopathies (AFM) via TRANSLAMUSCLE (PROJECT 19507 and 22946), Agence Nationale pour la Recherche (ANR) grant Epimuscle (ANR 11 BSV2 017 02) and RHU CARMMA (ANR-15-RHUS-0003). This work was also supported by Labex REVIVE (ANR-10-LABX-73), including a post-doctoral grant to JEdL and a PhD fellowship to L.M.

## Author contributions

J.E.d.L. designed, performed, and analyzed the majority of the experiments. R.B.A., L.M., and Y.L. performed and analyzed experiments, contributing equally to this work. B.D.L. performed experiments. F.J.D. supervised experiments. F.R. designed and supervised experiments and oversaw the project. J.E.d.L. and F.R. wrote the manuscript. R.B.A., L.M., Y.L., B.D.L., and F.J.D. read and edited the manuscript.

## Competing interests

The authors declare no competing interests.
