## [Peer Review File · Nature Communications]

Reviewers' Comments:

Reviewer #1:

Remarks to the Author:

In this manuscript the authors have performed analysis of conditional HIRA deficiency in satellite cells, by histological analysis of regenerative potential of Pax7CreErt2;Hirafl/fl mice, and parallel genome-wide transcriptional and epigenetic analysis of clones of C2C12 cells in which Hira knockdown was achieved by Crispr/Cas9-based genome editing. The authors show that Hira deficient satellite cells fail to regenerate and self-renew, leading to tissue repair failure. Hira deficiency in proliferating C2C12 cells prevents the activation of muscle genes, while upregulating genes from other lineages. Epigenetic analysis of Hira deficient C2C12 cells showed a reduction in H3.3 deposition and H3K27ac modification at regulatory regions of muscle genes, while expression of genes from alternative lineages in Hira-deficient C2C12 cells was associated with MLL1/MLL2-mediated increase of H3K4me3 mark their promoters.

The results are interesting and might reveal epigenetic features of one of the most fascinating and important issue in adult stem cell biology – that is, the epigenetic regulation of cell identity and ability to commit toward specific lineages. Moreover, the data generated are high-quality and well analyzed. Below general and specific issues that I suggest to address in order to improve this manuscript.

In general, as it stands, the manuscript remains more descriptive than providing mechanistic insights; this might not be a problem, as long as data generated are coherently referred to the most fundamental biological process here – the requirement of Hira-mediated deposition H3.3 histone variant in determining the myogenic identity of satellite cells. In this regard, it is important that the data generated in satellite cells and C2C12 are cross-validated. The authors should also better clarify whether H3.3 deposition is implicated in the maintenance of muscle stem cell identity in quiescent satellite cells, and/or in determining their myogenic identity during commitment while proliferating following activation in response to muscle injury.

Specific points:

1 - RNAseq and ChIPseq analysis is only performed in Hira deficient C2C12 cells. However, culture conditions are not well specified, except for RNAseq, which was performed in proliferating C2C12. Assuming the also ChIPseq data refer to the same conditions, this raises concerns on why most of the downregulated genes and related histone modifications refer to muscle differentiation, given that the differentiation process was not induced.

2 - C2C12 might not be truly representative of satellite cells. They are cultured cell lines that have been already committed to the myogenic lineage, express MyoD and Myf5 proteins and do not express Pax7 or other stemness genes. In these cells Hira knockdown is expected to affect C2C12 proliferation and/or differentiation (when induced to differentiate). Conversely, Pax7-driven Hira deficiency in satellite cells is expected to affect muscle stem cells at a stage (the muscle stem cell stage) that precedes their myogenic commitment. In this regard, it would have been more consistent to use MyoD- or Myf5-driven deficiency of Hira in satellite cells, as follow up of the data shown in Fig. 1. Regardless, as the works stands currently, the results need to be cross validated between these two cell models used by the authors. As the largest amount of data have been generated in C2C12 myoblasts, RNAseq should be easily performed in satellite cells and ChIPqPCR could be used for validation of representative genes.

3 - One global conclusion from the data shown could be that Hira-mediated deposition of H3.3 is required for both satellite cell commitment to the myogenic lineage (and possibly generation of Pax7 expressing satellite cells by asymmetric division) and maintenance of the myogenic lineage in already committed C2C12 cells. The authors should either experimentally validate/challenge these conclusions or at least provide an insightful discussion.

4 - Data shown implies that H3.3 incorporation into nucleosomes is required for increased histone acetylation at promoter of muscle genes and their expression, but not for expression of genes from alternative lineages, which is otherwise associated with H3K4me3 at promoters. Again, I suggest that the authors provide mechanistic insights into this different activation of gene expression (muscle vs other lineage genes) in satellite cells.

Reviewer is Pier Lorenzo Puri

Reviewer #2:

Remarks to the Author:

de Lima et coworkers report that the histone H3.3 chaperone HIRA regulates myogenesis and prevents expression of non-myogenic lineage genes.

Genomic deletion of HIRA in myogenic C2C12 cells resulted in reduced expression of Pax7 and myogenic genes and upregulation of alternative lineage genes.

Similarly, inducible deletion of HIRA in muscle stem cells (MuSCs) of injured mice resulted in decreased Pax7 expression and impaired regeneration.

H3.3 ChIP-seq in HIRA knock-out (KO) C2C12 cells revealed 10-fold reduction in the H3.3 genomic deposition compared to control cells. H3.3 was observed to be specifically enriched at the TSS of downregulated genes in HIRA KO cells. However, the H3.3 ChIP-seq signal was also overall decreased. In contrast, H3K27acetylation (ac) was not on average altered; increased H3K27ac was observed at upregulated and decreased at downregulated genes. H3K27ac was also reduced at sites with decreased ATAC-seq signal in C2C12 HIRA KO cells.

Expression of the H3K4 methyltransferases MLL1/2 was increased in C2C12 HIRA KO cells and resulted in upregulation of non-myogenic lineage genes. While ChIP-seq H3K4me3 signal was overall increased, H3K27me3 was unaffected in C2C12 HIRA KO cells. The Hoxb cluster, transcriptionally silent in C2C12 cells, displayed increased H3K4me3 and H3K27me3 loss. Other cell lineage-specific genes, such as the adipogenic Sox6 and Cebpa, had increased H3K4me3 signal at their promoters but no H3K27me3 changes upon HIRA deletion. Other neuronal, endothelial, mesenchymal, and osteogenic genes acquired H3K4me3 in HIRA- deleted C2C12. Deletion of MLL1/2 reduced expression of aberrantly expressed genes in double HIRA-MLL1 and MLL2 C2C12 cells.

This manuscript reports and mechanistically extends observations regarding the role of H3.3 in myogenesis. The observations are interesting and the conclusions supported by the data presented.

COMMENTS:

1. Despite a pervasive H3.3 distribution at promoters and gene bodies (Figure 3), there is no general decrease in either H3K27ac or H3K27me3 in HIRA KO C2C12 cells. Are K27 acetylation and methylation provided by canonical H3.1/.2 histones?
2. Were non-myogenic genes upregulated in quiescent or activated MuSCs of HIRA KO mice? Was any of the corresponding proteins detected in regenerating muscles?
3. Figure 4C. Were the transcripts of non-myogenic genes upregulated in HIRA KO C2C12 cells?
4. MLL1 is required for Pax7 expression. Was Pax7 affected in HIRA-MLL1/2 KO C2C12 cells?

Reviewer #4:

Remarks to the Author:

This investigation centres on the role of HIRA in the epigenetic specification and maintenance of muscle gene expression attributes in myogenic precursor cells, via its role in specific targeting of H3.3 to muscle specific genes. The approach is mainly by targeted KO of HIRA in the C2C12 myogenic cell line, backed up by conditional KO in mouse satellite cells followed by precipitation of an episode of muscle regeneration.

The molecular biological aspect of the work is convincingly followed up by intensive comparison of gene expression in C2C12 cells with and without HIRA KO, showing a clear drop in expression of myogenic genes in the latter together with an increased expression of genes characteristic of other mesenchymal cell lineages. This is followed up in detail in tissue culture.

The manuscript is clearly written, with few idiomatic faults and the conclusions are of fundamental interest to the topic of stem cell lineage commitment but the following points should be addressed.

In vivo experiments confirm a reduced regeneration in muscles of conditional HIRA-KO mice but this is a somewhat muted effect. Satellite cell numbers in regenerating muscle are reduced but are reported as numbers per fibre, which, in muscle exhibiting reduced regeneration, is a moving denominator, making interpretation difficult – some reference to muscle weight/size and fibres per unit area are needed to provide any real basis for comparison.

The data on regeneration includes the increase in prevalence of small fibres and the and the higher frequency of fibres expression developmental myosin isoforms in HIRA-KO muscle, is also difficult to interpret, Since the KO affects expression of muscle genes, it is uncertain whether these effects are attributable to its effect on the satellite cells or the muscle fibres to which they contribute.

Overall, the manuscript makes a good case for the heavy implication of HIRA in the stabilization of the myogenic phenotype via its selective activation of muscle genes. However, I think there should be some discussion of whether this selectivity is intrinsic to this particular chaperone protein or whether it resides to any extent in other factors.

Reviewer comments and responses

We thank the editor and reviewers for positive and constructive comments on the manuscript. We addressed all the reviewer concerns, performed additional experiments, clarified the manuscript, and reorganized the text and figures according to the editor and reviewer comments. We describe below, point by point the changes we made to address the points raised. In summary, regarding new experimental data we performed the following:

- RNA-seq in satellite cells FACS-isolated after muscle injury in control and cKO (new Fig. 4a-e)
- RT-qPCR in satellite cells FACS-isolated after muscle injury (new Fig. 4f)
- ChIP-RT-qPCR for H3.3 in satellite cells FACS-isolated after muscle injury (new Fig. 4g)
- ChIP-RT-qPCR for H3ac (pan-acetyl) in C2C12 control and *Hira* KO (new Fig. 5k)
- ChIP-RT-qPCR for H3.3 and H3.1 in C2C12 control and *Hira* KO (new Supplementary Fig. 2e, f)
- Immunostaining for VE-Cadherin (*Cdh5*) in satellite cells (new Supplementary Fig. 3i, j)
- Quantification of PAX7 and M-Cadherin cell number in 7dpi muscles, normalized on the area (new Supplementary Fig. 3c-e)
- RT-qPCR analysis of *Pax7* expression levels in C2C12, *Hira* KO, *Hira/Mll1* and *Hira/Mll2* dKO cell lines (new Supplementary Fig. 4i)

Reviewer #1 (Remarks to the Author):

In this manuscript the authors have performed analysis of conditional HIRA deficiency in satellite cells, by histological analysis of regenerative potential of *Pax7CreErt2;Hira^{fl/fl}* mice, and parallel genome-wide transcriptional and epigenetic analysis of clones of C2C12 cells in which *Hira* knockdown was achieved by Crispr/Cas9-based genome editing. The authors show that *Hira* deficient satellite cells fail to regenerate and self-renew, leading to tissue repair failure. *Hira* deficiency in proliferating C2C12 cells prevents the activation of muscle genes, while upregulating genes from other lineages. Epigenetic analysis of *Hira* deficient C2C12 cells showed a reduction in H3.3 deposition and H3K27ac modification at regulatory regions of muscle genes, while expression of genes from alternative lineages in *Hira*-deficient C2C12 cells was associated with MLL1/MLL2-mediated increase of H3K4me3 mark their promoters.

The results are interesting and might reveal epigenetic features of one of the most fascinating and important issue in adult stem cell biology – that is, the epigenetic regulation of cell identity and ability to commit toward specific lineages. Moreover, the data generated are high-quality and well analyzed. Below general and specific issues that I suggest to address in order to improve this manuscript.

In general, as it stands, the manuscript remains more descriptive than providing mechanistic insights; this might not be a problem, as long as data generated are coherently referred to the most fundamental biological process here – the requirement of *Hira*-mediated deposition H3.3 histone variant in determining the myogenic identity of satellite cells. In this regard, it is important that the data generated in satellite cells and C2C12 are cross-validated. The authors should also better clarify whether H3.3 deposition is implicated in the maintenance of muscle stem cell identity in quiescent satellite cells, and/or in determining their myogenic identity during commitment while proliferating following activation in response to muscle injury.

We appreciate the reviewer insightful comments and agree that additional cross-validation between the two cellular systems analyzed would improve our study. We performed the requested experiments and clarified in the text (in particular in the discussion) the mechanism of H3.3 incorporation at myogenic loci. According to our data, H3.3 incorporation at myogenic loci is associated with both commitment and maintenance of myogenic fate.

Specific points:

1 - RNAseq and ChIPseq analysis is only performed in Hira deficient C2C12 cells. However, culture conditions are not well specified, except for RNAseq, which was performed in proliferating C2C12. Assuming the also ChIPseq data refer to the same conditions, this raises concerns on why most of the downregulated genes and related histone modifications refer to muscle differentiation, given that the differentiation process was not induced.

We clarified this point in the material and methods. We also clarified the details of the culture conditions of the cells before collecting them for ChIP-seq in the text as requested, notably in the section describing the results of Figure 1. The cells were kept in culture at low density in high serum conditions to maintain their proliferative status and avoid terminal differentiation. C2C12 cells were cultured under identical conditions before being collected for RNA-seq, ChIP-seq and ATAC-seq to allow comparison of the results from the three experiments.

Regarding the second point, we agree that some of the dysregulated genes are associated with differentiation (like Myh genes, which are expressed at low levels in proliferating C2C12 and the protein is not detected at this point) but we observe that most of the myogenic dysregulated genes are actually associated with muscle commitment rather than differentiation, including genes related to 1) myogenesis (Figure 1k and 1l; Myf5, Myod1); 2) embryonic muscle development (Figure 1i, 1k and 1l; Fgfr4, Fgf2, Eya4, Dmrt2) and 3) satellite cell markers (Figure 1k and 1l; Pax7, Ncam1, Col5a1 and Col5a2, Heyl). Consistently, the downregulated GO terms relate to muscle tissue morphogenesis and development rather than terminal differentiation or fusion (Figure 1i).

2 – C2C12 might not be truly representative of satellite cells. They are cultured cell lines that have been already committed to the myogenic lineage, express MyoD and Myf5 proteins and do not express Pax7 or other stemness genes. In these cells Hira knockdown is expected to affect C2C12 proliferation and/or differentiation (when induced to differentiate). Conversely, Pax7-driven Hira deficiency in satellite cells is expected to affect muscle stem cells at a stage (the muscle stem cell stage) that precedes their myogenic commitment. In this regard, it would have been more consistent to use MyoD- or Myf5-driven deficiency of Hira in satellite cells, as follow up of the data shown in Fig. 1. Regardless, as the works stands currently, the results need to be cross validated between these two cell models used by the authors. As the largest amount of data have been generated in C2C12 myoblasts, RNAseq should be easily performed in satellite cells and ChiPqPCR could be used for validation of representative genes.

We thank the reviewer for this relevant question. We agree that C2C12 cells are not representative of satellite cells and we did not intend or claimed to use C2C12 cells as a model for satellite cells. C2C12 cells were used in our study as an adult myogenic cell line that allowed easy access to a large number of cells required for epigenetic studies. While we included data on the role of HIRA in adult muscle satellite cells using conditional mutant mice and regeneration studies, we do agree that a stronger cross-validation with satellite cells would improve our study. According to the reviewer suggestions we performed the following experiments:

1) We performed RNA-seq on injured (5dpi, where the number of satellite cells is maximum) and FACS-isolated satellite cells from controls and cKO animals. In addition, in order to obtain a more homogeneous population within the 5dpi satellite cell pool, we cultured the cells for 48h (to allow activation of all the cells). The results for this analysis are shown in the new Figure 4 (Fig. 4a-e). We further validated by RT-qPCR some of the dysregulated genes identified in the RNA-seq of satellite cells (Fig. 4f).

2) We performed ChIP-RT-qPCR for H3.3 in 5dpi FACS-isolated satellite cells and analyzed the loci of some of the downregulated myogenic genes (new Fig. 4g).

3 - One global conclusion from the data shown could be that Hira-mediated deposition of H3.3 is required for both satellite cell commitment to the myogenic lineage (and possibly generation of Pax7 expressing satellite cells by asymmetric division) and maintenance of the myogenic lineage in already committed C2C12 cells. The authors should either experimentally validate/challenge these conclusions or at least provide an insightful discussion.

We thank the reviewer for raising this point. Indeed, the fact that the satellite cell numbers are decreased when *Hira* is deleted but the proliferation of PAX7-positive cells is not affected suggests that asymmetric cell division, required for satellite cell pool replenishment, might be abrogated. Although, since in our *Hira* cKO model we lose satellite cell numbers, addressing asymmetric cell division that requires crossing with complex genetic mouse models based on different Cre drivers (Myf5-Cre while we have to use Pax7CreERT2) is technically impossible. As suggested by the reviewer, we addressed this point in the discussion. We further added to the discussion insights on the commitment versus maintenance conclusions from our data.

4 - Data shown implies that H3.3 incorporation into nucleosomes is required for increased histone acetylation at promoter of muscle genes and their expression, but not for expression of genes from alternative lineages, which is otherwise associated with H3K4me3 at promoters. Again, I suggest that the authors provide mechanistic insights into this different activation of gene expression (muscle vs other lineage genes) in satellite cells.

The H3K27ac mark is enriched at the promoter regions of upregulated genes in *Hira* KO, although this is not associated with changes in H3.3 incorporation (please refer to Fig. 2c,d). This suggests that either H3.3 already present at these sites or other H3 histone variants can possibly be associated with the H3K27ac enrichment.

Acetylation can occur in other residues of the N-terminal tail of the H3 histone, which can possibly contribute to the activation of the expression of alternative lineage genes in *Hira* KO cells. In order to experimentally address this point, we performed ChIP-RT-qPCR using a H3ac pan-acetyl antibody (recognizes all acetylation occurring at H3 N-terminal tail) in control and *Hira* KO cells. We tested whether H3 acetylation was present in the promoters of the upregulated alternative lineage genes *Cdh5*, *Nefl* and *Hoxb13*, that we had already confirmed to be enriched for H3K4me3 (Fig. 5j). We observed that at the promoters of these genes H3ac is significantly increased in *Hira* KO compared to control cells (new Fig. 5k). This suggests that increased H3K27ac and increased acetylation in other H3 residues at promoter regions of alternative lineage genes in *Hira* KO cells leads to the activation of their expression.

Reviewer #2 (Remarks to the Author):

de Lima et coworkers report that the histone H3.3 chaperone HIRA regulates myogenesis and prevents expression of non-myogenic lineage genes. Genomic deletion of HIRA in myogenic C2C12 cells resulted in reduced expression of Pax7 and myogenic genes and upregulation of

alternative lineage genes. Similarly, inducible deletion of HIRA in muscle stem cells (MuSCs) of injured mice resulted in decreased Pax7 expression and impaired regeneration. H3.3 ChIP-seq in HIRA knock-out (KO) C2C12 cells revealed 10-fold reduction in the H3.3. genomic deposition compared to control cells. H3.3 was observed to be specifically enriched at the TSS of downregulated genes in HIRA KO cells. However, the H3.3 ChIP-seq signal was also overall decreased. In contrast, H3K27acetylation (ac) was not on average altered; increased H3K27ac was observed at upregulated and decreased at downregulated genes. H3K27ac was also reduced at sites with decreased ATAC-seq signal in C2C12 HIRA KO cells.

Expression of the H3K4 methyltransferases MLL1/2 was increased in C2C12 HIRA KO cells and resulted in upregulation of non-myogenic lineage genes. While ChIP-seq H3K4me3 signal was overall increased, H3K27me3 was unaffected in C2C12 HIRA KO cells. The Hoxb cluster, transcriptionally silent in C2C12 cells, displayed increased H3K4me3 and H3K27me3 loss.

Other cell lineage-specific genes, such as the adipogenic Sox6 and Cebpa, had increased H3K4me3 signal at their promoters but no H3K27me3 changes upon HIRA deletion. Other neuronal, endothelial, mesenchymal, and osteogenic genes acquired H3K4me3 in HIRA-deleted C2C12. Deletion of MLL1/2 reduced expression of aberrantly expressed genes in double HIRA-MLL1 and MLL2 C2C12 cells.

This manuscript reports and mechanistically extends observations regarding the role of H3.3 in myogenesis. The observations are interesting and the conclusions supported by the data presented.

We thank the reviewer for considering the importance and relevance of our work and for the positive comments.

COMMENTS:

1. Despite a pervasive H3.3 distribution at promoters and gene bodies (Figure 3), there is no general decrease in either H3K27ac or H3K27me3 in HIRA KO C2C12 cells. Are K27 acetylation and methylation provided by canonical H3.1/2 histones?

We have now performed ChIP-RT-qPCR on *Hira* cells to examine changes in H3.3 and H3.1 enrichment at multiple target genes. We observe that upon loss of HIRA, H3.3 is lost at the *Ccnd1* and *Gapdh* promoters, but this loss is compensated by an increased enrichment of histone H3.1 (Supplementary Fig. 2e, f). Mass spectrometry studies have shown that H3K27me3 marks are highly represented on histone H3.1, and therefore we believe that maintenance of global H3K27me3 levels upon *Hira* KO is likely due to the mark being present on histone H3.1. With respect to the H3K27ac marks, we do observe loss of this mark at specific promoters which coincides with loss of histone H3.3. While global levels of H3K27ac are not decreased at a general level, we expect that this mark is happening on histone H3.1. However, confirmation of this would require the development of antibodies that can differentiate marks at the lysine 27 position between different histone H3 variants. To highlight the replacement of H3.3 by H3.1, we have now added a sentence to the text during the discussion of the reduced accessibility as observed by ATAC-Seq (results describing Figure 2 and Supplementary Fig. 2).

2. Were non-myogenic genes upregulated in quiescent or activated MuSCs of HIRA KO mice? Was any of the corresponding proteins detected in regenerating muscles?

We analyzed the gene expression of some non-myogenic genes in activated satellite cells (FACS-isolated after injury) and we confirmed that activated satellite cells lacking *Hira* display increased expression of non-myogenic genes (new Fig. 4f). We also collected activated satellite

cells and performed immunostainings. We confirmed the presence of VE-Cadherin (*Cdh5* gene) protein in cKO cells compared to control (new Supplementary Fig. 3i,j).

3. Figure 4C. Were the transcripts of non-myogenic genes upregulated in HIRA KO C2C12 cells?

We confirmed that non-myogenic genes are upregulated in Hira KO C2C12 cells, data is shown in Figure 1k and 1l.

4. MLL1 is required for Pax7 expression. Was Pax7 affected in HIRA-MLL1/2 KO C2C12 cells?

We addressed the expression levels of Pax7 according to the reviewer suggestion. Pax7 expression is downregulated in Hira KO and deleting Mll1 or Mll2 in the Hira KO cell line increases Pax7 expression levels but does not rescue the expression to control levels. These data are now included in the new Supplementary Fig. 4i.

Reviewer #4 (Remarks to the Author):

This investigation centres on the role of HIRA in the epigenetic specification and maintenance of muscle gene expression attributes in myogenic precursor cells, via its role in specific targeting of H3.3 to muscle specific genes. The approach is mainly by targeted KO of HIRA in the C2C12 myogenic cell line, backed up by conditional KO in mouse satellite cells followed by precipitation of an episode of muscle regeneration. The molecular biological aspect of the work is convincingly followed up by intensive comparison of gene expression in C2C12 cells with and without HIRA KO, showing a clear drop in expression of myogenic genes in the latter together with an increased expression of genes characteristic of other mesenchymal cell lineages. This is followed up in detail in tissue culture.

The manuscript is clearly written, with few idiomatic faults and the conclusions are of fundamental interest to the topic of stem cell lineage commitment but the following points should be addressed.

We thank the reviewer for considering our work to be of fundamental interest to the lineage commitment field. We addressed below each of the points raised by the reviewer.

In vivo experiments confirm a reduced regeneration in muscles of conditional HIRA-KO mice but this is a somewhat muted effect. Satellite cell numbers in regenerating muscle are reduced but are reported as numbers per fibre, which, in muscle exhibiting reduced regeneration, is a moving denominator, making interpretation difficult – some reference to muscle weight/size and fibres per unit area are needed to provide any real basis for comparison.

We agree with the reviewer that a normalization of the cell number per area can be more accurate in the context of regeneration failure. We therefore added cell quantification graphs normalized per unit area (new Supplementary Fig. 3c-e).

The data on regeneration includes the increase in prevalence of small fibres and the and the higher frequency of fibres expression developmental myosin isoforms in HIRA-KO muscle, is also difficult to interpret, Since the KO affects expression of muscle genes, it is uncertain whether these effects are attributable to its effect on the satellite cells or the muscle fibres to which they contribute.

As correctly stated by the reviewer, myogenic gene expression is decreased in cKO cells, which could be linked with the direct effect on muscle satellite cells. While genes associated with myogenic commitment are systematically decreased (Fig. 1k,l), the different myosin heavy chain genes are either upregulated or downregulated (Fig. 1k), which is probably linked to impaired regeneration and changes in the fiber type. However, even if the expression of embryonic myosin heavy chain (Myh3) is downregulated (Fig. 1k), this might not reflect an abrogation at the protein level. In addition, we believe that the defective satellite cell epigenetic landscape is leading to defective myoblasts that will eventually lead to fusion defects, as opposed to a specific effect on the fibers.

Overall, the manuscript makes a good case for the heavy implication of HIRA in the stabilization of the myogenic phenotype via its selective activation of muscle genes. However, I think there should be some discussion of whether this selectivity is intrinsic to this particular chaperone protein or whether it resides to any extent in other factors.

We thank the reviewer for allowing us to clarify this point. In our study we demonstrate that HIRA confers an appropriate epigenetic landscape that promotes muscle gene expression via H3.3 incorporation, which is required to maintain the active transcription-associated mark H3K37ac, at myogenic loci. However, whether the H3.3 deposition-associated regulation of myogenic gene expression is an intrinsic function of the histone chaperone HIRA or if other factors are involved remains to be determined. This point is now addressed in depth in the discussion.

Reviewers' Comments:

Reviewer #1:

Remarks to the Author:

The authors have satisfactorily addressed all reviewer comments and the manuscript is definitely further improved and suitable for publication.

Reviewer #2:

Remarks to the Author:

The authors have satisfactorily addressed my comments.

Reviewer #4:

Remarks to the Author:

The authors have largely dealt with my criticisms of the original paper and the current version presents a convincing overall case for the main thesis in this paper from the in vitro experimental work, where the notion of myogenic cell 'identity' given a descriptive reality in terms of a number of markers.

The in vivo work is less convincing because the functional outcome of the conditional HIRA- KO on muscle regeneration are not firmly established as criteria of myogenic cell identity, due in good part to the pleiotropic effects of this gene that are only partly revealed by the tissue culture data. For instance, although muscle fibre size and myosin isoform expression are widely used as indicators of defects in function of myogenic cells, they are weak criteria because these phenotypes are attributable not only to the number of nuclei inserted into the fibres by fusion of the myogenic cells but also by the performance of the resulting myonuclei within the fibres in terms of expression of muscle proteins. The latter does not obviously fall within the definition of identity of the myogenic cells and should be discussed. The lower satellite cell density in regenerating `HIRA=KO muscle provide stronger evidence of a disturbance of myogenic cell function, but as timepoint snapshots of a highly dynamic process, the numbers are only around 1 fold apart, which, if considered as a manifestation of proliferation, is only one round of cell division apart, and does not present a watertight case for a loss of myogenic identity in so complex a process. The work presented in this manuscript is on the leading edge of technical excellence but its limitations as well as its clearer messages should be given full exposure.

REVIEWERS' COMMENTS

We thank the reviewers for positive comments and support with the final revisions.

Reviewer #1 (Remarks to the Author):

The authors have satisfactorily addressed all reviewer comments and the manuscript is definitely further improved and suitable for publication.

Reviewer #2 (Remarks to the Author):

The authors have satisfactorily addressed my comments.

Reviewer #4 (Remarks to the Author):

The authors have largely dealt with my criticisms of the original paper and the current version presents a convincing overall case for the main thesis in this paper from the in vitro experimental work, where the notion of myogenic cell 'identity' given a descriptive reality in terms of a number of markers.

The in vivo work is less convincing because the functional outcome of the conditional HIRA- KO on muscle regeneration are not firmly established as criteria of myogenic cell identity, due in good part to the pleiotropic effects of this gene that are only partly revealed by the tissue culture data.

For instance, although muscle fibre size and myosin isoform expression are widely used as indicators of defects in function of myogenic cells, they are weak criteria because these phenotypes are attributable not only to the number of nuclei inserted into the fibres by fusion of the myogenic cells but also by the performance of the resulting myonuclei within the fibres in terms of expression of muscle proteins. The latter does not obviously fall within the definition of identity of the myogenic cells and should be discussed. The lower satellite cell density in regenerating `HIRA=KO muscle provide stronger evidence of a disturbance of myogenic cell function, but as timepoint snapshots of a highly dynamic process, the numbers are only around 1 fold apart, which, if considered as a manifestation of proliferation, is only one round of cell division apart, and does not present a watertight case for a loss of myogenic identity in so complex a process. The work presented in this manuscript is on the leading edge of technical excellence but its limitations as well as its clearer messages should be given full exposure.

We thank the reviewer for finding that our data is convincing and improved with the revision. We agree with the reviewer concern regarding the in vivo results where the observed regeneration failure could be associated with a myofiber/myonuclei phenotype rather than a myogenic identity maintenance of the satellite cells. We believe that given the consistency between the in vivo and in vitro data it is possible that satellite cell altered transcriptome could consequently affect a myofiber phenotype. Although we did not address this experimentally we addressed this point in the "Discussion" and highlighted some perspectives on how to overcome this issue.